# Causal analysis of Covid-19 spread in Germany

**Atalanti A. Mastakouri**
Department of Empirical Inference
Max Planck Institute for
Intelligent Systems
Tübingen, Germany
atalanti.mastakouri@tuebingen.mpg.de

**Bernhard Schölkopf**
Department of Empirical Inference
Max Planck Institute for
Intelligent Systems
Tübingen, Germany
bs@tuebingen.mpg.de

## Abstract

In this work, we study the causal relations among German regions in terms of the spread of Covid-19 since the beginning of the pandemic, taking into account the restriction policies that were applied by the different federal states. We loose a strictly formulated assumption for a causal feature selection method for time series data, robust to latent confounders, which we subsequently apply on Covid-19 case numbers. We present findings about the spread of the virus in Germany and the causal impact of restriction measures, discussing the role of various policies in containing the spread. Since our results are based on rather limited target time series (only the numbers of reported cases), care should be exercised in interpreting them. However, it is encouraging that already such limited data seems to contain causal signals. This suggests that as more data becomes available, our causal approach may contribute towards meaningful causal analysis of political interventions on the development of Covid-19, and thus also towards the development of rational and data-driven methodologies for choosing interventions.

## 1   Introduction

The ongoing outbreak of the Covid-19 pandemic has rendered the tracking of the virus spread a problem of major importance, in order to better understand the role of the demographics and of non pharmaceutical interventions (NPIs) made to contain the virus. Until 15/5/2020, 175,699 cases and 8,001 deaths were recorded in Germany, a country with a population of 83 million people, 16 federal states with independent local governments, and 412 districts (Landkreise). In this paper, we focus on a causal time series analysis of the Covid-19 spread in Germany, aiming to understand the spatial spread and the causal role of the applied NPIs.

Causal inference from time series is a fundamental problem in data science, and many papers provide solutions for parts of the problem subject to necessary assumptions [1–6]. The main difficulty in this research problem is the possibility of hidden confounding in the data, as it is almost impossible in real datasets to have observed all the necessary information. Another problem is a characteristic of the time series themselves, which create dependencies due to connections in the past, hindering the formulation of necessary d-separation statements for graphical inference [7]. Finally, many known methods cannot handle instantaneous effects that may exist among the time series.

We consider a problem with small sample size compared to the dimension of its covariates, yet of significant current importance: the tracking of the spread of the Covid-19 pandemic, based only on the reported cases. Not having access to all relevant covariates and to all

interventions that were applied at different times by different regions constitutes a heavily confounded problem, whose causal analysis requires a method which is robust to hidden confounders. Tracking the Covid-19 spread is of interest since it may help understand and contain the virus. There are significant efforts to understand this based on individual location or proximity information [8]. Other efforts try to understand and quantify the importance of applied NPIs through modelling of the spread [9–12]. In the present work, we focus on the causal analysis of the spread. We perform an offline causal inference analysis of the reported daily Covid-19 case numbers in regions of Germany, in combination with the NPIs that were made to contain the spread.

The most common and established approach for causal inference on time-series is Granger causality [13–15]. In the multivariate case, we say that $X^j$ *Granger-causes* $X^k$ ($k \neq j$) if a conditional dependence $X^k_t \not\perp\!\!\!\perp X^j_{\text{past}(t)} \mid \mathbf{X}^{-j}_{\text{past}(t)}$ exists (here, $-j$ denotes all indices other than $j$, and past($t$) denotes all indices $t' < t$). The fundamental disadvantage of this method is its reliance on *causal sufficiency*: the assumption that all the common causes in the system are observed; in other words, that no hidden confounders can exist [16]. Violations of this, common in real world data, render Granger causality and its extensions [e.g. 17, 18] incorrect, yielding misleading conclusions.

Below, we loose an unecessarily strictly phrased assumption of the SyPI algorithm [6], a causal feature selection method for time series with latent confounders. We apply it on Covid-19 cases reported by German regions, with the goal to detect which regions and which restriction policies played a causal role on the formation and modulation of the regionally-reported daily cases. We perform this analysis on a state and on a district level. We compare our findings with predictions of the widely used Lasso-Granger [19] and tsFCI method [21], showing that SyPI yields more meaningful results. Note that while no ground truth exists, our detected causes tend to be neighbouring states/regions, with discrepancies that can often plausibly be attributed to the existence of major transportation hubs.

## 2 Methods and Tasks

### 2.1 Causal inference on time series

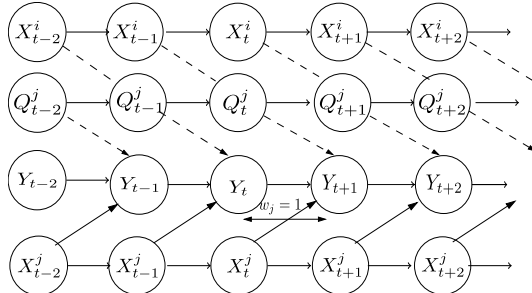

Figure 1: An example full time graph of two observed $(X^i, X^j)$, one potentially hidden $(Q^j)$ and one target $(Y)$ time series.

For the problem of causal feature selection on time series data, we are given observations from a target time series $Y := (Y_t)_{t \in \mathbb{Z}}$ whose causes we wish to find, and observations from a multivariate time series $\mathbf{X} := ((X^1_t, \ldots, X^d_t))_{t \in \mathbb{Z}}$ of potential causes (candidate time series). In settings were Causal sufficiency cannot be assumed, like the one we tackle here, unobserved multivariate time series, which may act as common causes of the observed ones also exist. An example of such a setting is given in Figure 1.

### 2.2 The SyPI method

Here we use the SyPI method proposed by [6], as it can give causal conclusions in large, dense graphs of time series, based solely on observational data and without assuming causal

sufficiency. According to [6], the method requires, as input, observations from a target time series $Y$ and from a multivariate time series (candidate causes) $\mathbf{X}$, as defined above. Moreover, it allows for unobserved multivariate time series, which may act as hidden confounders. Under suitable assumptions discussed in Section 2.3, the method provably detects all the direct causes of the target and some indirect ones, but never confounded ones (its conditions are both necessary and sufficient[1]).

We now try to provide some intuition. Since it requires familiarity with (and terminology of) causal structure learning, some readers may want to consult [6]. For each candidate causal time series $X^i$ that has a dependency with the target $Y$ at lag $w_i$, the method performs targeted isolation of the path $X^i_{t-1} \rightarrow X^i_t \text{ - -} Q^j_{t'} \dashrightarrow Y_{t+w_i}$ (where $w_i \in Z, t' < t + w_i, Q^j \in \mathbf{X}^{-\mathbf{i}}$ or unobserved), that contains, for every candidate $i$, the current $X^i_t$ and the previous time step $X^i_{t-1}$ of the candidate causal time series, and the corresponding node of the target time series $Y_{t+w_i}$.[2] It does so by building a conditioning set that contains the nodes of $\mathbf{X}^{-\mathbf{i}}$ that enter node $Y_{t+w_i-1}$ (temporal ancestor of the target node $Y_{t+w_i}$ of the same time series), including the node itself. This way, it exploits the fact that if there is a confounding path between $X^i_t$ and $Y_{t+w_i}$, then $X^i_t$ will be a collider that will unblock the path between $X^i_{t-1}$ and $Y_{t+w_i}$ when we condition on it. Therefore, running SyPI boils down to testing two conditions: condition 1 examines if $X^i_t$ and $Y_{t+w_i}$ are conditionally dependent given the aforementioned conditioning set, and condition 2 examines if $X^i_{t-1}$ and $Y_{t+w_i}$ become conditionally independent if $X^i_t$ is included in the aforementioned conditioning set. If both conditions hold true, then SyPI identifies $X^i$ as a cause of $Y$ [6].

## 2.3 Weakening SyPI's assumptions

According to [6] SyPI is a sound and complete causal feature selection method in the presence of latent common causes subject to certain graph restrictions. Among the most important graphical assumptions required is that the target be a sink node (assumption 6 in [6]), i.e., the target has no descendants. In Theorem A below, we relax this strictly phrased assumption, proving that it suffices that none of the (direct or indirect) descendants of the target belongs in the pool of the candidate causes. While this relaxation is important for our application, we prefer not to repeat all assumptions and definitions from [6]. Rather, we describe below what needs to be adapted to handle our more general setting.

The intuition behind Theorem A is the following. The original assumption 6 ensures that when an unconfounded path $X^i_t \rightarrow Y_{t+w_i}$ for some lag $w_i$ exists, the true cause $X^i$ will not be rejected due to a parallel path $X^i_t \rightarrow X^j_{t'} \leftarrow Y_{t+w_i}$ that contains a collider $X^j_{t'}$, which could potentially be unblocked rendering condition 2 of Theorem 2 in [6] false. Theorem 1 of [6] remains unaffected from whether $Y$ is a sink node or not, because in the case that it is not, i.e., $X^i_t \leftarrow Y_{t+w_i}$, condition 2 will correctly reject $X^i_t$.

Therefore, we only need to show that Theorem 2 of [6] remains unaffected if instead of $Y$ being a sink node, all of its descendants do not belong in $\mathbf{X}$ (we write $\mathbf{DE}^{\mathcal{G}}_Y \notin \mathbf{X}$). In the case that all the descendants of $Y$ do not belong in its candidate causes $\mathbf{X}$, then they will be unobserved. Assume there is one descendant $D \notin \mathbf{X}$ of $Y$ that is also connected with a node $X^i_t$ from $\mathbf{X}$. Then $D$ can only have incoming arrows from $X^i_t$ and therefore $D$ is an unobserved collider (any out-coming arrow from $D$ to $\mathbf{X}$ will violate the assumption $\mathbf{DE}^{\mathcal{G}}_Y \notin \mathbf{X}$). Therefore any path that contains the unobserved collider $D$ cannot be unblocked to create any additional dependencies, because $D$ and any of its descendants cannot belong in the conditioning set.

**Theorem A** (Theorems 1 and 2 from [6] still apply). *Given the target time series $Y$ and the candidate causes $\mathbf{X}$, assuming Causal Markov condition, causal faithfulness, no backward arrows in time $X^i_{t'} \nrightarrow X^j_t, \forall t' > t, \forall i, j$, stationarity of the full time graph as well as assumption A7-A9 from [6], if the target $Y$ is not a sink node, but, instead, none of its*

*descendants belongs in* $\mathbf{X}$*:* $\mathbf{DE}_Y^{\mathcal{G}} \notin \mathbf{X}$*, then Theorem 1 and 2 from [6] still apply. That means the conditions of Theorem 1 from [6] are still sufficient for identifying direct and indirect causes, and conditions of Theorem 2 from [6] are still necessary for identifying all the direct unconfounded causes in single-lag dependency graphs.*

We prove Theorem A in the Appendix (Section 9.2.2). Moreover, in Figure 6 of the Appendix we provide some simulated experiments ran on 100 random graphs for varying number of observed time series, taking into account the modified assumption.

While the relaxed assumption makes the result more generally applicable, we need one additional step to apply it to our dataset: The algorithm requires as input the candidates and the target as two separate variables. Therefore, we need to assign one region at a time as **target**. In order to comply with the aforementioned assumption, instead of directly feeding all the remaining time series as the candidate causes of the target $Y$, we use as **candidate causes** those other regions that **have reported Covid-19 cases before the target** (in addition to the applied policies for the analysis at the federal states level). This makes it more likely that no effects of the target exist in its candidate causes (assuming stationarity of the graph).

SyPI assumes that the causal relations among the time series are stationary, not changing in different time windows. However, since we do not know the ground truth, it is possible that the policies not only cause the reported infections time series but also be caused by it in different time windows. This possible violation of stationarity of the graph creates problems because it also implies arrows from the target to some of the policies time series which belong in its candidate causes. Therefore this could violate both the assumption 6 in [6] about the target being a sink node and the relaxed proposed assumption $\mathbf{DE}_Y^{\mathcal{G}} \notin \mathbf{X}$. We are aware that this could happen, which is one reason we are careful in our conclusions.

### 2.4 Selection of statistical thresholds

Since the causal Markov condition and causal faithfulness are assumed (definitions 9.2.1, 9.2.1), there is an equivalence between d-separation statements in the graph and conditional independences on the probability distributions of the variables. As [6], we use SyPI for linear relationships only (although the theory is more general), and hence resort to partial correlations to test the conditional dependence (condition 1) and the conditional independence (condition 2) of [6]. SyPI operates with two thresholds for those two tests: one for rejecting conditional independencies (condition 1), and another for accepting conditional independencies (condition 2). Since the time series of the daily reported cases since the beginning of the pandemic in Germany include only 87 reported days (until 15/05/2020), we decided to explore the outcomes of the algorithm for stricter and looser thresholds. We thus examined values of threshold-1 in $\{0.01, 0.05\}$ and values for threshold-2 in $\{0.1, 0.2\}$. We report the causal findings for the looser combination $(0.05, 0.1)$ in Fig. 3a and for all four in the App. Fig. 5.

## 3 Experiments

### 3.1 Dataset: Daily reported Covid-19 cases for German regions

The data are taken from the official reports of the Robert-Koch Institute, last downloaded on 15/05/2020 [20]. They are analysed in two steps:

**Causal analysis on federal state level**  Figure 2 depicts daily reported Covid-19 cases for each of the 16 German federal states, each one represented by a time series, starting from when the first report was made (28/01/2020) until 15/05/2020. The plots are sorted chronologically, with the top left corresponding to the Bundesland (federal state) that reported first, and the bottom right the Bundesland that reported Covid-19 cases last. In addition, we created indicator functions for nine NPIs that were imposed separately in each Bundesland, as gathered from the official German states' websites and from `https://calc.systemli.org/u0o26ims15cr`. The periods these measures were in effect are depicted as indicator functions (vertically scaled to make sure all are visible) in the above plots. The policies

are: *closing of schools, closing of universities, ban of gatherings of more than 1000 people, ban of gatherings of more than 10 people, obligatory quarantine of 14 days after returning from risk areas, ban of gatherings of more than 2 people, closing restaurants, closing hotels, forbidding visits in hospitals and nursing homes.* We provide the data in the supplement and here `https://owncloud.tuebingen.mpg.de/index.php/s/r4dPdpSBAzP6Ee5`. Note that not all policies were applied in all federal states, and also that for the state of Niedersachsen, no policies are provided. We apply the algorithm for each target state independently, keeping as candidates all the federal states that have reported cases before the target one, as well as the nine aforementioned policies for the specific target. Results are shown in Figure 3a. The datasets were re-evaluated, as well, after four months (until 26/09/2020). The updated time series, the data as well as the results are provided in Figure 10 and in Section 9.6 in the Appendix.

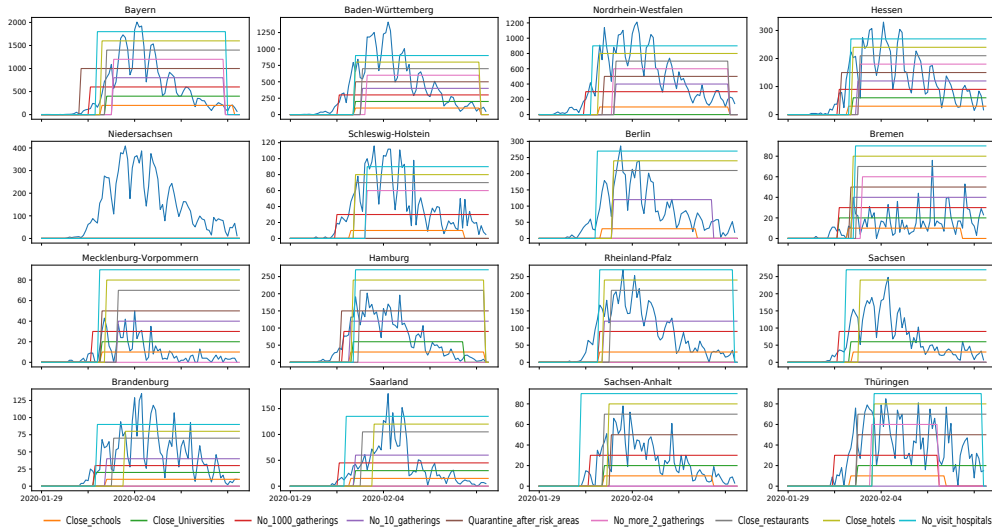

Figure 2: Time series of daily reported detected Covid-19 cases in each federal state from 28/01/2020 until 15/05/2020. The blue curve represents the daily reported infections as a function of time. In addition to the Covid-19 cases, 9 restriction measures are depicted as indicator functions (see legend and main text). The height of the indicator functions does not have a meaning. It is only adjusted for visibility purposes.

**Causal analysis on district level** To get further results, we apply the modified SyPI method on the time series of daily reported Covid-19 cases for all 412 districts of Germany. We apply it the following way: For every district we use SyPI twice; the first time using as candidate causes all the neighbouring districts of the target that have reported cases before, and the second time, using the same number of districts but from random non-neighbour (distant) locations that have also reported cases before the target. Our hope would be that SyPI will identify more causes among the neighbour districts than among the non-neighbour ones. Furthermore, we would hope that (some of) the latter could be justified by a large airport close-by. The default thresholds of SyPI $(0.01, 0.2)$ were used. For this analysis, we created a matrix with all the neighbour districts of each district, as well as the location of the largest airports (including the number of flights from the past years), which we also provide in the supplement. Furthermore, for the largest airports in terms of number of passengers per year, according to the German flight security organisation (DFS) (MUC, STR, TXL, FDH, FMM, NUE, HAM, FRA, HHN, HAJ, NRN, CGN, DUC, DMT, DRS, BRE, KSF, SCN), we check which districts are near (within 40km) each one of these. In total, 169 out of the total 412 districts were found to be near one of the large airports. The 40km distance was chosen as it corresponds to the diameter of a medium size German district. We then categorise our results in four categories: 1. Detected causes among the neighbours of the target, 2. Detected causes near (within 40km) the target, 3. Detected causes near (within 40km) a large airport, 4. Distant targets that cannot be categorised otherwise.

## 3.2 Comparison against Lasso-Granger and tsFCI

We compare the SyPI method for the spread of Covid-19 in the German federal states, with Lasso-Granger [19] and tsFCI [21]. Granger causality is the most widely used method for causal time series analysis, although it assumes causal sufficiency, which we expect to be heavily violated in real data. The main difference between the proposed approach and tsFCI is that SyPI pre-calculates a very concise conditioning set for each target and only requires two conditional independence (CI) tests per candidate cause, to decide if it is a true cause of the target. In contrast, tsFCI performs exhaustively CI tests for all possible combinations of conditioning sets and lags, which results in very ambiguous statistical results and very large computational times in large graphs. Of course, tsFCI aims at the full graph discovery and not only at causal feature selection. This also justifies tsFCI's more computationally intensive conditions. For fair comparison we used the same threshold for all the statistical tests of both methods (0.05). The policies were not included in this analysis, because tsFCI algorithm cannot handle at the same time both binary and continuous data.

## 4 Results

### 4.1 SyPI on Covid-19 cases and policies in the federal states

In Figure 3a, the policies and federal states that were identified as causes by SyPI are depicted with target/color specific arrows. Fig. 3a correspond to the "looser" combination of thresholds $(0.05, 0.1)$. Figure 5 (supplement) provides results for all four combinations. As we can see, the result does not change dramatically with different threshold combinations, but as expected, more causes are detected with the "looser" combination $(0.05, 0.1)$, as it more easily accepts dependencies and independencies. We discuss the findings in Section 5.1.

### 4.2 Enacted policies and causal roles of federal states

Here we discuss the relation between the outcome of the above causal analysis and the applied NPIs until 15/05/2020. This time, instead of looking for causes of Covid-19 cases in German federal states, we look at the states that helped contain the spread of the pandemic by not causing others. We make an observation that may serve as additional sanity check about the causal predictions of SyPI, using its stricter thresholds results:[3] states that were not found to cause other states were those that closed schools and universities "early enough" (meaning before 100 cases were reported): Bremen, Thüringen, Saarland, Brandenburg, Sachsen-Anhalt and Sachsen (with the exception of Mecklenburg-Vorpommern). In addition, the German states that were found to cause others were also those that either did not take both measures combined (Schleswig-Holstein, Nordrhein-Westfalen, Rheinland-Pfalz [4]), or they took them relatively late (i.e., > 100 cases) (Bayern, Baden-Württemberg, Hamburg, Hessen).

### 4.3 Causal spread of Covid-19 among the German districts

Since the number of federal states is relatively small, we ran our analysis also at a finer level of granularity, using districts rather than states. Figure 3b depicts the map of Germany with all the detected causal districts for each district. Arrows with solid lines show neighbour causes, while arrows with dashed lines depict causes that do not share a border with the target. We see that for the majority of the target districts the detected causes are neighbouring districts, and that those that are not are generally near a large airport or within 40km distance from the target. Note that since the dashed arrows are significantly longer than the solid ones, the Fig. 3b at first glance seems to show mostly dashed arrows. This is misleading; for a numerical comparison, see Figure 4a. Distant causal districts often seem to be aligned with the routes of the domestic connections with the highest traffic, as reported in the DFS's latest flight report [22].

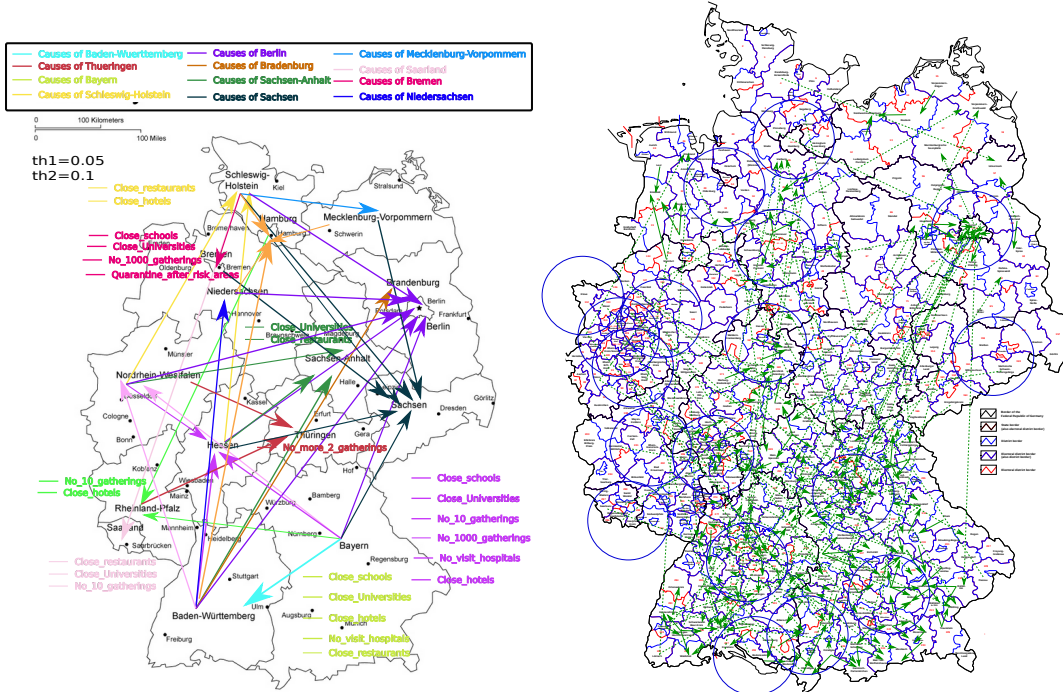

(a) Federal state level & policies causal analysis.  (b) District-level causal analysis.

Figure 3: a) Detected causal paths of the spread of Covid-19 among the German federal states, including causes among the NPIs taken by each federal state. Each colour (in arrows and policies) indicates causes of one state (see top legend). These findings correspond to the looser of the four combinations of thresholds $(0.05, 0.1)$ that we tested. Results for the remaining three combinations can be found in Fig. 5 in the Appendix. b) Detected causal districts for the spread of Covid-19, for each district, using the modified SyPI algorithm. Solid arrows depict causes that are neighbour districts (i.e., sharing a common border). Dashed arrows depict causes that are not. The majority of the detected non-neighbour causes are close to cities with larger airports (MUC, STR, TXL, FDH, FMM, NUE, HAM, FRA, HHN, HAJ, NRN, CGN, DUC, DMT, DRS, BRE, KSF, SCN), and the majority of the detected causes are neighbours to the target. Note that since the dashed arrows are significantly longer than the solid ones, the Figure at first glance seems to show mostly dashed arrows. This is misleading; for a numeric comparison, see Figure 4a. Blue cycles indicate 40km radius around the largest airports. For the district-level analysis, the default thresholds of SyPI were used $(0.01, 0.2)$.

These are: Berlin - Munich, Berlin - Frankfurt, Düsseldorf - Munich, Cologne/ Bonn - Munich, Düsseldorf - Berlin, Stuttgart - Hamburg, Frankfurt - Munich, Berlin - Stuttgart, with 1-2 Mi flights per year. These paths can be seen in Fig. 3b, as detected longer-range dashed causal arrows. Table 2 in the Appendix includes all the causal results shown in Fig. 3b. We categorise the total number of 231 causes detected into the following four categories: 1. Detected causes neighbouring (sharing common borders) the target district, 2. Detected causes near ($\sim$ 40km) to the target, 3. Detected causes close to a large airport, 4. Distant targets that cannot be categorised otherwise. As we can see in Fig. 4a, the majority of causes are neighbour districts, and only the 12% of the causes cannot be justified by proximity to the target or a large airport. Fig. 4b shows the histogram of the detected causes that are located close to a large airport, in cases where also the target is reachable by another airport.

## 4.4 SyPI vs Lasso-Granger and tsFCI

Table 1 in the supplement presents all the detected causes (states and policies) of each federal state, for the SyPI method and thresholds $(0.05, 0.1)$, as well as for the Lasso-Granger

method. With these "loose" thresholds we expect the largest number of detected causes with SyPI (corresponding to the bottom left map in Figure 3a). We see that Lasso-Granger detects almost all candidates as causes. This indicates that this is a dataset with many latent confounders, which forces Granger to give incorrect causal claims. On the other hand, SyPI is robust against false positives due to latent common causes, and thus gives potentially more meaningful results. Figure 7 in the Appendix depicts the detected causes (federal states) of each federal state for SyPI (thresholds $0.05, 0.05$) and tsFCI (threshold $0.05$). As we can see, tsFCI detected eight, while SyPI 44 directed edges (causes). Four of the detected causes by tsFCI were a subset of the ones detected by SyPI. For the majority of the remaining states tsFCI yielded '↔', without being able to conclude to one direction. SyPI needed only 19 seconds to run, while tsFCI needed 15 minutes for the same dataset.

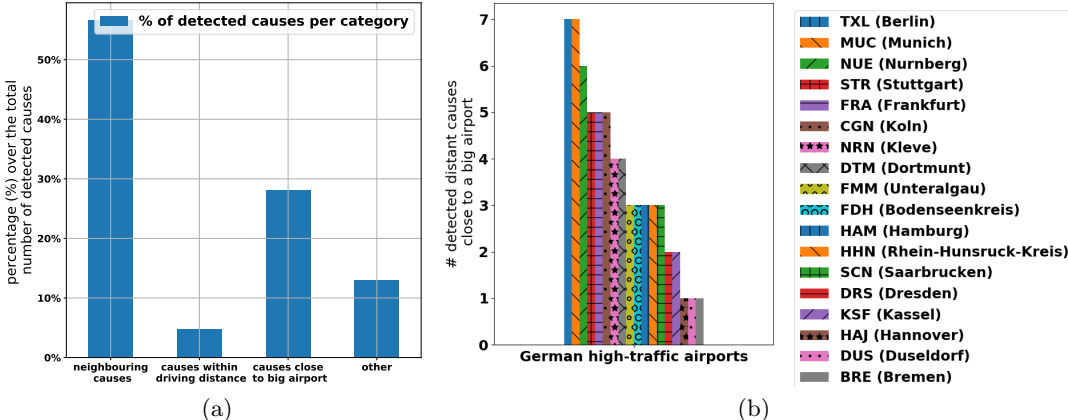

(a)  (b)

Figure 4: a) Percentage of detected district causes (out of 231 detected causes in total) by category of district relative to target district. Most causes of a given target turn out to be neighbouring districts, and of the distant ones, many are close to major airports. Only 12% of the detected causes cannot be justified by proximity to the target or to a larger airport. b) Detected distant causes located close to the large airports. To assign a detected causes to one of the airports it had to meet two criteria: 1. located in near the airport (see main text), and 2. the target of this cause also needs to be located close to another big airport. We sort the airports by the number of detected causes.

## 5 Discussion

### 5.1 Findings of the causal analysis

We performed a causal analysis both on a federal state/policy level, and on a more fine-grained district level. We tried normalising the case numbers in various ways (e.g., dividing by the maximum), but the results were not much affected. We decided not to normalise the data by population, as we felt this would unduly enhance the influence of less populous states. For the policy analysis, we compared the findings of SyPI with the predictions of the widely used Lasso-Granger [19] as well as tsFCI [21]. We did not compare with seqICP [2], which is a feature selection method, since this method requires that no interventions are applied on the target. In the present setting, the target always is subject to interventions. Lasso-Granger detected almost every candidate region and distancing measure as causal, which is not surprising in a confounded real-world dataset like the present one. On the contrary, tsFCI detected very few directed arrows.

SyPI, on the other hand, yielded more meaningful results. We saw that the causes detected by SyPI on a district level tended to be neighbouring German districts, modulo the presence of major airports (which tend to be associated with industrial hubs). The pattern for federal states was consistent with this, but since there are fewer federal states than districts, numbers are small. The results in Figure 3a seem meaningful in that much of the spread is local. In addition, Bayern and Baden-Württemberg, the federal states with the largest current

Covid-19 incidences,[5] have almost no arrows coming in from other states (see also Fig. 5). In the district-level analysis only 38 out of 167 detected causes of targets in these two federal states belonged to another state. The majority of detected causes was due to internal mobility (84.7% for Baden-Württemberg and 73% for Bayern). This finding is also in line with mobility charts in Germany that show a gradual increase from April on, mostly on the states of Baden-Württemberg, Bayern and Berlin [6]. It is believed that those states (which lie in the South) had a strong influx of cases from Italy and Austria, where the pandemic took hold earlier.[7] As a noteworthy detail, our algorithm identified Tirschenreuth (northeast Bavaria) as the cause of all its neighbouring districts (Wunsiedel im Fichtelgebirge, Bayreuth, Neustadt an der Waldnaab). On March 7th, a large festival took place in Mitterteich in the district of Tirschenreuth, with a strong subsequent local COVID-19 outbreak.[8] Furthermore, we saw that for the federal states, different restriction policies were found as causal, yet the analysis until the 15/05/2020 showed that for the majority of states the closing of the universities and schools was detected as causal. Our findings about the causal role of banning gatherings of more than 1000 people, followed by closing of schools and ban of meetings of more than 2 people, are also in agreement with the modeling analysis of [9].

## 5.2 Validity of assumptions made

A potential issue is the time delay between the application of a restriction measure and the observation of its effects on the target. Note that schools and universities were (often) closed later than some other measures were taken, e.g., the ban on larger gatherings. With SyPI, it is hard to infer which policy had the strongest effect unless we knew exactly the incubation time of each measure and actually shift the time series of cases by a corresponding amount. As we learn more about epidemiological parameters of Covid-19 (e.g., the typical time delay between infection and being tested positive), we may be able to perform the latter analysis.

As mentioned in Section 2.3, we assume that the policies affect the target (Covid-19 cases) and not the other way around, in order to comply with our requirement that none of the descendants of the target belongs in its candidate causes. This may be violated if policies were adjusted based on the observed number of positive cases. We can be sure from the theoretical point of view that the detected causes are not confounded covariates. However, the method will likely have failed to detect all the true causes, if the aforementioned violation applies. With Theorem A we relaxed the strictly phrased assumption of SyPI about the target being a sink node, by requiring only that it has no descendants among its candidates. In practice, we try to ensure this by selecting as candidates only regions that have already reported cases before the target. This makes it likely that no (or few) effects of the target exist in its candidate causes.

## 5.3 Contributions & conclusions

Motivated by an application on Covid-19 spreading, we relaxed a strictly phrased assumption of the causal feature selection algorithm SyPI of [6], making it applicable to a causal analysis of daily reported Covid-19 cases of German states and districts, and state-wise social distancing measures. While ground truth is not available, our results as discussed in Section 5.1 seem meaningful. Possibly the biggest weakness of our approach lies in the fact that the data we used is confined: (1) we only look at case numbers, in contrast to more sophisticated methods to track the spread of an epidemic using contact tracing or even genetic analyses [23]. Moreover, (2) the sample size is small (the pandemic still be relatively new), and (3) the political interventions considered are binary and thus also provide relatively little information. It is encouraging, however, that already such limited data seems to contain causal signals pertaining to a highly non-trivial task. This suggests that our approach may contribute towards meaningful causal analysis of political interventions on the spread of Covid-19 as more data becomes available.

## 6   Broader Impact

The causal analysis proposed in this paper aims at contributing to the broader effort of scientists to understand the spread of the Covid-19 pandemic and the causal role of political interventions such as social distancing. The causal method being applied and assayed in this work can provide trustworthy causal results, since it is robust against false positives in the presence of latent confounders in time series — note that in Covid-19 data science problems, with our limited present understanding, it is likely that relevant covariates are unobserved, leading to confounded problems. Despite the theoretical validity of the causal method, caution should be exercised in the interpretation of the results of the present study, due to the limited data available for this analysis (only daily reported Covid-19 cases for different regions, and some political interventions), and the sheer difficulty of the task. **At present, we would thus not recommend that our empirical findings be used to guide public policy.** However, we find our results encouraging, given the hardness of causal structure learning from observational real-world data, known to practitioners in the field [1]. We therefore believe that methods such as the one used above, and further developments based upon it, can contribute towards rational approaches for choosing and balancing restriction measures for pandemics such as Covid-19.

## 7   Funding disclosure

The work in this paper was supported and fully funded by Max Planck Institute for Intelligent Systems. The authors declare no conflict of interest.

## 8   Acknowledgements

The authors would like to thank Alexander Ecker and his team. The political intervention data during the early months of the pandemic was collected upon Alexander Ecker's initiative by Christina Thöne and a team of volunteers at `http://crowdfightcovid19.org`. Thanks also go to Karin Bierig for helping with the creation of a database of neighbouring districts and airports and for collecting information about the status of the NPIs the later months of the pandemic. Finally thanks go to Dominik Janzing, Vincent Stimper, Simon Buchholz and Julius von Kügelgen for their helpful feedback on the manuscript.

## Footnotes

[1]Although SyPI's conditions are necessary only for single-lag dependencies, the method has provided satisfying results even with multiple lags [6]. The existence of multiple lags would only result in fewer detected causes, without affecting the validity of the method in terms of false positives.

[2]'$\dashrightarrow$' denotes a directed path, '- -' denotes a collider-free path.

[3]Notice that this result should be treated with caution as it depends on the correctness of the above causal analysis and it may be confounded by the time order that the states reported causes.

[4]The arrow Rheinland-Pfalz → Thüringen does not appear in the subplot of strict thresholds because the p-value (0.011) for condition 1 was on the limit over the strict threshold 1 (0.01).

[5] https://www.rki.de/DE/Content/InfAZ/N/Neuartiges_Coronavirus/Situationsberichte/2020-05-31-en.pdf?__blob=publicationFile

[6] https://www.covid-19-mobility.org/mobility-monitor/

[7] https://www.rki.de/DE/Content/InfAZ/N/Neuartiges_Coronavirus/Situationsberichte/2020-03-13-en.pdf?__blob=publicationFile

[8] https://medicalxpress.com/news/2020-03-bavarian-town-germany-impose-full.html

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
