[Supplementary Material]

# 9 Appendix/ Supplementary material for the paper: Causal analysis of Covid-19 spread in Germany

## 9.1 Results of causal analysis on federal level for all four combinations of thresholds for SyPI

Figure 5: Detected causal paths of the spread of Covid-19 among the federal German states, including causes among the restriction measures taken by each federal state. Each colour (in arrows and policies) indicates causes of one state (see top legend). The four subfigures correspond to the four combinations of threshold 1 and 2 that we tested.

## 9.2 Theory

### 9.2.1 Definitions

**Definition** (Causal Faithfulness). *A distribution $P$ is faithful to a directed acyclic graph (DAG) $G$ if no conditional independence relations other than the ones entailed by the Markov property are present.*

**Definition** (Causal Markov Condition [16]). *Let $G$ be a causal graph with vertex set $\mathcal{V}$ and $P$ be a probability distribution over the vertices in $V$ generated by the causal structure represented by $G$. $G$ and $P$ satisfy the Causal Markov Condition if and only if for every $W$ in $\mathcal{V}$, $W$ is independent of $\mathcal{V} \setminus (Descendants(W) \cup Parents(W))$ given $Parents(W)$.*

*Here we use the global version of the Markov condition, which reads: if $\mathcal{X} \perp\!\!\!\perp_G \mathcal{Y} \mid \mathcal{Z} \Rightarrow \mathcal{X} \perp\!\!\!\perp \mathcal{Y} \mid \mathcal{Z}$ for all disjoint vertex sets $\mathcal{X}, \mathcal{Y}, \mathcal{Z}$ (where $\perp\!\!\!\perp_G$ denotes d-separation, as defined above)*

### 9.2.2 Proof of Theorem A

*Proof.* The proof of Theorem 1 in [6] applies without changes. Regarding Theorem 2 in [6]: Assume that the direct path $X_t^i \to Y_{t+w_i}$ exists and it is unconfounded. Then, condition 1 of Theorem 2 in [6] is true. Now assume that condition 2 of Theorem 2 in [6] does not hold. This would mean that the set $\{\boldsymbol{S^i}, X_t^i, Y_{t+w_i-1}\}$ does not d-separate $X_{t-1}^i$ and $Y_{t+w_i}$. (Recall that a path $p$ is said to be *d-separated* by a set of nodes in $Z$ if and only if $p$ contains a chain or a fork such that the middle node is in $Z$, or if $p$ contains a collider such that neither the middle node nor any of its descendants are in the $Z$.) Hence, a violation of condition 2 would imply that (a) there is some middle node or descendant of a collider in $\{\boldsymbol{S^i}, X_t^i, Y_{t+w_i-1}\}$ and no non-collider node in this path belongs to this set, or (b) that there is a collider-free path between $X_{t-1}^i$ and $Y_{t+w_i}$ that does not contain any node in $\{\boldsymbol{S^i}, X_t^i, Y_{t+w_i-1}\}$.

(a) *There is some middle node or descendant of a collider in $\{\boldsymbol{S^i}, X_t^i, Y_{t+w_i-1}\}$ and no non-collider node in this path belongs to this set:* the proof given in [6] remains unaffected if all $\mathbf{DE}_Y^{\mathcal{G}} \notin \mathbf{X}$, because any collider $D$ or descendent of collider between some $X_t^j$ and $Y_{t+w_i}$ will be unobserved, therefore will not be possible to belong in the conditioning set $\{\boldsymbol{S^i}, X_t^i, Y_{t+w_i-1}\}$.

(b) *There is a collider-free path between $X_{t-1}^i$ and $Y_{t+w_i}$ that does not contain any node in $\{\boldsymbol{S^i}, X_t^i, Y_{t+w_i-1}\}$:* the proof given in [6] remains unaffected.

$\square$

## 9.3 Simulation experiments

We applied SyPI with the modified assumption on simulated full time graphs, with varying numbers of observed time series and one hidden time series, for varying levels of noise. Dashed lines correspond to the missed direct causes false negative rate and the continuous lines to the false positive rate. We see that the result of the method remains unaffected by the relaxation of the assumption that we made. As expected from the proof of the theorem, FPR and FNR are not affected by the relaxation of the sink node assumption if $\mathbf{DE}_Y^{\mathcal{G}} \notin \mathbf{X}$.

Figure 6: False positive and false negative rates for missed direct causes, in simulated graphs with varying number of observed time series and one hidden, for varying noise levels. Dashed lines correspond to the missed direct causes false negative rate and the continuous lines to the false positive rate. We see that the result of the method remains unaffected by the relaxation of the assumption that we made. As expected from the proof of the theorem, FPR and FNR are not affected by the relaxation of the sink node assumption if $\mathbf{DE}_Y^{\mathcal{G}} \notin \mathbf{X}$.

## 9.4 Detailed findings from comparison of modified SyPI with Lasso-Granger and tsFCI of the Covid-19 spread among the German federal states.

Figure 7: Detected causes for each federal state until 15/05/2020. Left: SyPI (0.05, 0.05). Right: tsFCI (0.05). tsFCI detected eight, while SyPI 44 directed edges (causes). Four of the detected causes by tsFCI were a subset of the ones detected by SyPI. For the majority of the remaining states tsFCI yielded '↔', without being able to conclude to one direction.

Table 1: Detected causes for each federal German state (Bundesland), using SyPI with the loose combination of thresholds $(0.05, 0.1)$ (2nd column) and using Lasso-Granger (3rd column). As expected we see that the number of detected causes by Granger is multiple times more than those of SyPI; in most cases Granger detects as causes all the candidate states. Without knowing the ground truth, this is an obvious indication that the dataset includes hidden confounders, that make the federal states look all related to each other. This violation of causal sufficiency makes Granger to fail, as expected. On the other hand, SyPI does not suffer from such problems even when there are latent confounders.

| Target | Predicted causes by SyPI | Predicted causes by Granger |
|---|---|---|
| Bayern | Close schools, Close universities, Close restaurants, Close hotels, Prohibit visits at hospitals | Close schools, Close universities, No more than 1000 people gatherings, No more than 10 people gatherings, Quarantine 14 days after visiting risk areas, No more than 2 people gatherings, Close restaurants, Close hotels, Prohibit visits at hospitals |
| Baden-Württemberg | Bayern | Bayern, Close schools, Close universities, No more than 1000 people gatherings, No more than 10 people gatherings, Quarantine 14 days after visiting risk areas, No more than 2 people gatherings, Close restaurants, Close hotels, Prohibit visits at hospitals |
| Nordrhein-Westfalen | Bayern, Baden-Württemberg | Bayern, Baden-Württemberg, Close schools, No more than 1000 people gatherings, No more than 10 people gatherings, Quarantine 14 days after visiting risk areas, No more than 2 people gatherings, Close restaurants, Close hotels, Prohibit visits at hospitals |
| Hessen | Bayern, Baden-Württemberg, Nordrhein-Westfalen, Close schools, Close universities, No more than 1000 people gatherings, No more than 10 people gatherings, Close hotels, Prohibit visits at hospitals | Bayern, Baden-Württemberg, Nordrhein-Westfalen, Close schools, Close universities, No more than 1000 people gatherings, No more than 10 people gatherings, Quarantine 14 days after visiting risk areas, No more than 2 people gatherings, Close restaurants, Close hotels, Prohibit visits at hospitals |
| Niedersachsen | Baden-Württemberg | Bayern, Baden-Württemberg, Nordrhein-Westfalen, Hessen |
| Schleswig-Holstein | Nordrhein-Westfalen, Hessen, Close restaurants, Close hotels | Bayern, Baden-Württemberg, Nordrhein-Westfalen, Hessen, Niedersachsen, Close schools, No more than 1000 people gatherings, No more than 2 people gatherings, Close restaurants, Close hotels, Prohibit visits at hospitals |
| Berlin | Bayern, Baden-Württemberg, Nordrhein-Westfalen, Hessen, Niedersachsen, Schleswig-Holstein | Bayern, Baden-Württemberg, Nordrhein-Westfalen, Hessen, Niedersachsen, Schleswig-Holstein, Close schools, No more than 10 people gatherings, Close restaurants, Close hotels, Prohibit visits at hospitals |
| Bremen | Schleswig-Holstein, Close schools, Close universities, No more than 1000 people gatherings, Quarantine 14 days after visiting risk areas | Bayern, Baden-Württemberg, Nordrhein-Westfalen, Hessen, Niedersachsen, Schleswig-Holstein, Berlin, Close schools, Close universities, No more than 1000 people gatherings, No more than 10 people gatherings, Quarantine 14 days after visiting risk areas, No more than 2 people gatherings, Close restaurants, Close hotels, Prohibit visits at hospitals |
| Mecklenburg-Vorpommern | Schleswig-Holstein | Bayern, Baden-Württemberg, Nordrhein-Westfalen, Hessen, Niedersachsen, Schleswig-Holstein, Berlin, Bremen, Close schools, Close universities, No more than 1000 people gatherings, No more than 10 people gatherings, Quarantine 14 days after visiting risk areas, Close restaurants, Close hotels, Prohibit visits at hospitals |
| Hamburg | Baden-Württemberg, Niedersachsen, Schleswig-Holstein, Mecklenburg-Vorpommern | Bayern, Baden-Württemberg, Nordrhein-Westfalen, Hessen, Niedersachsen, Schleswig-Holstein, Berlin, Bremen, Mecklenburg-Vorpommern, Close schools, Close universities, No more than 1000 people gatherings, No more than 10 people gatherings, Quarantine 14 days after visiting risk areas, Close restaurants, Close hotels, Prohibit visits at hospitals |
| Rheinland-Pfalz | Bayern, Hamburg, No more than 10 people gatherings, Close hotels | Bayern, Baden-Württemberg, Nordrhein-Westfalen, Hessen, Niedersachsen, Schleswig-Holstein, Berlin, Bremen, Mecklenburg-Vorpommern, Hamburg, Close schools, No more than 1000 people gatherings, No more than 10 people gatherings, Close restaurants, Close hotels, Prohibit visits at hospitals |
| Sachsen | Bayern, Hessen, Schleswig-Holstein, Bremen, Mecklenburg-Vorpommern, Hamburg | Bayern, Baden-Württemberg, Nordrhein-Westfalen, Hessen, Niedersachsen, Schleswig-Holstein, Berlin, Bremen, Mecklenburg-Vorpommern, Hamburg, Rheinland-Pfalz, Close schools, Close universities, No more than 1000 people gatherings, Close hotels, Prohibit visits at hospitals |
| Brandenburg | Baden-Württemberg | Bayern, Baden-Württemberg, Nordrhein-Westfalen, Hessen, Niedersachsen, Schleswig-Holstein, Berlin, Bremen, Mecklenburg-Vorpommern, Hamburg, Rheinland-Pfalz, Sachsen, Close schools, Close universities, No more than 1000 people gatherings, No more than 10 people gatherings, Close restaurants, Close hotels, Prohibit visits at hospitals |
| Saarland | Schleswig-Holstein, Close universities, No more than 10 people gatherings, Close restaurants | Bayern, Baden-Württemberg, Nordrhein-Westfalen, Hessen, Niedersachsen, Schleswig-Holstein, Berlin, Bremen, Mecklenburg-Vorpommern, Hamburg, Rheinland-Pfalz, Sachsen, Brandenburg, Close schools, Close universities, No more than 1000 people gatherings, No more than 10 people gatherings, Close restaurants, Close hotels, Prohibit visits at hospitals |
| Sachsen-Anhalt | Baden-Württemberg, Nordrhein-Westfalen, Hessen, Close universities, Close restaurants | Bayern, Baden-Württemberg, Nordrhein-Westfalen, Hessen, Niedersachsen, Schleswig-Holstein, Berlin, Bremen, Mecklenburg-Vorpommern, Hamburg, Rheinland-Pfalz, Sachsen, Brandenburg, Saarland, Close schools, Close universities, No more than 1000 people gatherings, Quarantine 14 days after visiting risk areas, Close restaurants, Close hotels, Prohibit visits at hospitals |
| Thüringen | Nordrhein-Westfalen, Rheinland-Pfalz, No more than 2 people gatherings | Bayern, Baden-Württemberg, Nordrhein-Westfalen, Hessen, Niedersachsen, Schleswig-Holstein, Berlin, Bremen, Mecklenburg-Vorpommern, Hamburg, Rheinland-Pfalz, Sachsen, Brandenburg, Saarland, Sachsen-Anhalt, Close schools, Close universities, No more than 1000 people gatherings, Quarantine 14 days after visiting risk areas, No more than 2 people gatherings, Close restaurants, Close hotels, Prohibit visits at hospitals |

## 9.5 Detailed findings from the causal analysis of the Covid-19 spread among the German district states.

Table 2: Detected causes for each district German state, using SyPI. In the first column, the target district state is reported. In the second column, the detected causes among the neighbouring districts are reported. Finally, in the third column, we report the detected distant causes. Strict thresholds (the default of SyPI method) are used for the analysis. As explained in Section 4.3 and in Figure 4a, the majority of detected district causes are neighbours of the targets, and the majority of the distant detected causes are located close to a big airport.

| Target distric state | Detected neighbouring causes | Detected distant causes |
|---|---|---|
| SK Gelsenkirchen | [] | [] |
| LK Landsberg a.Lech | [] | [] |
| LK Starnberg | [] | [] |
| LK Fürstenfeldbruck | [] | SK Gelsenkirchen |
| SK München | [] | [] |
| LK Traunstein | [] | [] |
| SK Delmenhorst | [] | [] |
| LK München | LK Starnberg | [] |
| LK Freising | LK München | [] |
| SK Köln | [] | [] |
| LK Lippe | [] | [] |
| LK Stormarn | [] | [] |
| LK Ravensburg | [] | [] |
| LK Göppingen | [] | [] |
| LK Tübingen | [] | [] |
| SK Freiburg i.Breisgau | [] | [] |
| LK Rottweil | [] | [] |
| LK Heinsberg | [] | [] |
| LK Breisgau-Hochschwarzwald | SK Freiburg i.Breisgau | [] |
| LK Böblingen | [] | LK München |
| SK Erlangen | [] | [] |
| LK Ludwigsburg | [] | SK Freiburg i.Breisgau |
| LK Viersen | [] | [] |
| StadtRegion Aachen | [] | [] |
| SK Kaiserslautern | [] | [] |
| LK Wesel | [] | [] |
| SK Hamburg | [] | [] |
| LK Märkischer Kreis | [] | [] |
| SK Fürth | [] | [] |
| LK Heilbronn | LK Ludwigsburg | [] |
| LK Ostalbkreis | [] | [] |
| LK Gießen | [] | [] |
| SK Bonn | [] | [] |
| LK Alb-Donau-Kreis | LK Göppingen | SK Fürth |
| LK Segeberg | [] | [] |
| LK Rhein-Neckar-Kreis | [] | LK Böblingen |
| SK Mönchengladbach | [] | [] |
| LK Ostallgäu | [] | [] |
| SK Lübeck | [] | [] |
| SK Schwabach | [] | [] |
| LK Lahn-Dill-Kreis | [] | [] |
| SK Bremen | [] | [] |
| SK Duisburg | [] | [] |
| LK Oberhavel | [] | [] |
| LK Düren | [] | [] |
| LK Groß-Gerau | [] | [] |
| SK Heilbronn | [] | [] |
| SK Münster | [] | [] |
| Region Hannover | [] | [] |
| LK Borken | [] | [] |
| SK Frankfurt am Main | [] | [] |
| LK Herzogtum Lauenburg | [] | [] |
| LK Hochtaunuskreis | [] | [] |
| LK Zollernalbkreis | LK Rottweil, LK Tübingen | [] |
| SK Nürnberg | SK Erlangen | LK Segeberg, SK Gelsenkirchen |
| LK Rheinisch-Bergischer Kreis | [] | [] |
| SK Mannheim | [] | [] |
| LK Rhein-Kreis Neuss | [] | [] |
| LK Sächsische Schweiz-Osterzgebirge | [] | [] |
| LK Ebersberg | LK München | [] |
| LK Cuxhaven | [] | [] |
| LK Rosenheim | LK Ebersberg, LK München | [] |
| SK Berlin Marzahn-Hellersdorf | [] | [] |
| SK Berlin Mitte | [] | [] |
| SK Berlin Neukölln | [] | [] |
| SK Ulm | [] | [] |
| LK Passau | [] | [] |
| LK Saale-Orla-Kreis | [] | [] |
| LK Lörrach | [] | [] |
| LK Rems-Murr-Kreis | LK Heilbronn, LK Ludwigsburg | LK Ebersberg, LK Wesel |
| LK Rhein-Sieg-Kreis | [] | [] |
| LK Main-Kinzig-Kreis | SK Frankfurt am Main | [] |
| LK Pinneberg | [] | [] |
| LK Esslingen | LK Rems-Murr-Kreis | [] |
| LK Bergstraße | [] | [] |
| LK Karlsruhe | [] | LK Freising |
| LK Oberbergischer Kreis | [] | [] |
| LK Ammerland | [] | [] |

| Target distric state | Detected neighbouring causes | Detected distant causes |
| --- | --- | --- |
| LK Vorpommern-Greifswald | [] | [] |
| SK Bochum | [] | [] |
| SK Berlin Tempelhof-Schöneberg | [] | [] |
| LK Rotenburg (Wümme) | [] | [] |
| LK Mecklenburgische Seenplatte | [] | SK Berlin Tempelhof-Schöneberg |
| LK Main-Tauber-Kreis | [] | [] |
| LK Coesfeld | [] | [] |
| SK Düsseldorf | [] | [] |
| SK Berlin Pankow | [] | SK Nürnberg |
| SK Stuttgart | [] | [] |
| LK Emmendingen | LK Breisgau-Hochschwarzwald, SK Freiburg i.Breisgau | [] |
| SK Berlin Friedrichshain-Kreuzberg | SK Berlin Mitte, SK Berlin Tempelhof-Schöneberg | LK Starnberg |
| LK Sigmaringen | [] | [] |
| LK Grafschaft Bentheim | [] | [] |
| SK Mainz | [] | [] |
| SK Heidelberg | SK Mannheim | [] |
| LK Bad Dürkheim | [] | [] |
| LK Germersheim | [] | [] |
| LK Neckar-Odenwald-Kreis | LK Heilbronn | LK Breisgau-Hochschwarzwald |
| LK Cham | [] | [] |
| SK Koblenz | [] | [] |
| SK Oldenburg | [] | [] |
| LK Leer | [] | [] |
| LK Aichach-Friedberg | [] | [] |
| LK Vorpommern-Rügen | [] | LK Zollernalbkreis, SK Münster |
| LK Roth | [] | [] |
| LK Bodenseekreis | LK Ravensburg | [] |
| LK Osnabrück | [] | [] |
| LK Stade | [] | [] |
| LK Rhein-Erft-Kreis | [] | [] |
| LK Rheingau-Taunus-Kreis | LK Hochtaunuskreis | [] |
| LK Neu-Ulm | SK Ulm | [] |
| LK Unna | [] | [] |
| LK Weilheim-Schongau | LK Starnberg | LK Viersen |
| LK Waldeck-Frankenberg | [] | [] |
| LK Oberallgäu | LK Ravensburg, LK Ostallgäu | [] |
| LK Vogelsbergkreis | LK Gießen | LK Borken |
| LK Ortenaukreis | LK Emmendingen | [] |
| SK Berlin Reinickendorf | [] | [] |
| LK Miesbach | LK Rosenheim | LK Sigmaringen |
| SK Braunschweig | [] | [] |
| LK Dithmarschen | [] | [] |
| LK Hohenlohekreis | [] | [] |
| SK Dortmund | [] | [] |
| LK Calw | LK Karlsruhe | LK Ravensburg |
| LK Bad Kissingen | [] | [] |
| LK Euskirchen | [] | [] |
| LK Celle | Region Hannover | [] |
| SK Würzburg | [] | [] |
| LK Erlangen-Höchstadt | SK Erlangen | LK Ammerland, SK Berlin Mitte |
| LK Havelland | [] | LK Ludwigsburg |
| LK Konstanz | LK Sigmaringen | [] |
| SK Ingolstadt | [] | [] |
| LK Würzburg | [] | LK Erlangen-Höchstadt |
| SK Karlsruhe | LK Karlsruhe | LK Lahn-Dill-Kreis, LK Bodenseekreis |
| SK Kempten | [] | [] |
| SK Leipzig | [] | [] |
| SK Augsburg | [] | [] |
| LK Biberach | LK Neu-Ulm | [] |
| LK Minden-Lübbecke | [] | [] |
| LK Bautzen | [] | [] |
| LK Mettmann | [] | [] |
| LK Harburg | SK Hamburg | SK Erlangen, LK Gießen, LK Zollernalbkreis |
| SK Berlin Charlottenburg-Wilmersdorf | [] | [] |
| SK Bielefeld | [] | [] |
| LK Herford | [] | [] |
| LK Kassel | [] | [] |
| SK Essen | [] | [] |
| SK Rosenheim | LK Rosenheim | [] |
| SK Hof | [] | [] |
| LK Warendorf | [] | [] |
| SK Wilhelmshaven | [] | [] |
| LK Rastatt | [] | [] |
| LK Bitburg-Prüm | [] | [] |
| LK Fürth | [] | [] |
| LK Enzkreis | [] | SK Ingolstadt |
| SK Dresden | [] | [] |
| SK Baden-Baden | [] | [] |
| LK Ennepe-Ruhr-Kreis | [] | [] |
| LK Hildesheim | [] | [] |
| LK Offenbach | SK Frankfurt am Main | [] |
| LK Steinfurt | [] | [] |
| LK Schwarzwald-Baar-Kreis | LK Breisgau-Hochschwarzwald | SK Berlin Reinickendorf, LK Rhein-Neckar-Kreis |
| SK Erfurt | [] | [] |
| LK Freudenstadt | LK Tübingen | LK Oberallgäu |
| LK Regensburg | LK Cham | LK Segeberg |
| LK Tuttlingen | LK Schwarzwald-Baar-Kreis, LK Sigmaringen, LK Zollernalbkreis | LK Germersheim, SK Koblenz, LK Vogelsbergkreis |

| Target distric state | Detected neighbouring causes | Detected distant causes |
| --- | --- | --- |
| LK Pfaffenhofen a.d.Ilm | LK Aichach-Friedberg, LK Freising | LK Bergstraße |
| LK Teltow-Fläming | SK Berlin Tempelhof-Schöneberg | [] |
| LK Schwandorf | LK Regensburg | LK Borken |
| LK Reutlingen | LK Esslingen | LK Lörrach, SK Freiburg i.Breisgau |
| LK Rostock | LK Vorpommern-Rügen | [] |
| LK Friesland | [] | LK Lörrach, LK Viersen |
| SK Aschaffenburg | [] | [] |
| SK Berlin Spandau | LK Havelland | LK Oberallgäu |
| LK Merzig-Wadern | [] | [] |
| LK Spree-Neiße | [] | [] |
| LK Saar-Pfalz-Kreis | [] | [] |
| SK Osnabrück | [] | [] |
| LK Schwäbisch Hall | LK Rems-Murr-Kreis | LK Ludwigsburg, SK Hof, SK Berlin Friedrichshain-Kreuzberg |
| LK Plön | [] | [] |
| LK Dingolfing-Landau | [] | [] |
| SK Offenbach | [] | [] |
| LK Dachau | [] | [] |
| LK Straubing-Bogen | [] | [] |
| LK Saarlouis | [] | [] |
| LK Stadtverband Saarbrücken | [] | [] |
| LK Rottal-Inn | [] | SK Berlin Charlottenburg-Wilmersdorf |
| SK Wiesbaden | [] | [] |
| SK Bottrop | [] | [] |
| LK Donau-Ries | LK Aichach-Friedberg | [] |
| LK Kelheim | LK Freising, LK Pfaffenhofen a.d.Ilm, LK Regensburg | LK Düren |
| LK Landshut | LK Dingolfing-Landau, LK Freising, LK Kelheim, LK Regensburg, LK Rottal-Inn | LK Vogelsbergkreis |
| SK Bremerhaven | [] | [] |
| LK Leipzig | [] | [] |
| SK Berlin Steglitz-Zehlendorf | [] | LK Stadtverband Saarbrücken, LK Pinneberg, LK Schwäbisch Hall |
| LK Lindau | LK Bodenseekreis | LK Kelheim |
| LK Main-Spessart | LK Bad Kissingen | SK Fürth, LK Lippe |
| LK Marburg-Biedenkopf | [] | [] |
| SK Berlin Lichtenberg | SK Berlin Marzahn-Hellersdorf | LK Fürth |
| SK Hagen | [] | [] |
| LK Görlitz | [] | [] |
| LK Garmisch-Partenkirchen | LK Ostallgäu, LK Weilheim-Schongau | SK Heidelberg |
| LK Fulda | LK Bad Kissingen | LK Sächsische Schweiz-Osterzgebirge |
| LK Neunkirchen | [] | [] |
| LK Mayen-Koblenz | [] | [] |
| LK Neuwied | [] | [] |
| LK Elbe-Elster | [] | [] |
| LK Emsland | LK Leer, LK Osnabrück, LK Steinfurt | [] |
| LK Oldenburg | [] | [] |
| LK Neustadt a.d.Aisch-Bad Windsheim | [] | LK Mayen-Koblenz |
| LK Erding | LK Freising, LK Landshut, LK München | [] |
| LK Oberspreewald-Lausitz | [] | [] |
| SK Pforzheim | [] | [] |
| SK Berlin Treptow-Köpenick | SK Berlin Friedrichshain-Kreuzberg | [] |
| SK Krefeld | [] | [] |
| LK Siegen-Wittgenstein | [] | [] |
| SK Kiel | [] | [] |
| LK Soest | [] | [] |
| LK Westerwaldkreis | [] | [] |
| SK Leverkusen | [] | [] |
| SK Chemnitz | [] | [] |
| SK Halle | [] | [] |
| SK Weimar | [] | [] |
| LK Waldshut | LK Breisgau-Hochschwarzwald | SK Nürnberg, LK Schwandorf |
| SK Weiden i.d.OPf. | [] | [] |
| LK Tirschenreuth | [] | [] |
| SK Solingen | [] | [] |
| SK Rostock | [] | [] |
| LK Vulkaneifel | [] | [] |
| SK Frankenthal | [] | [] |
| SK Magdeburg | [] | [] |
| SK Remscheid | [] | [] |
| LK Verden | [] | [] |
| SK Eisenach | [] | [] |
| LK Rhein-Hunsrück-Kreis | [] | [] |
| LK Paderborn | [] | [] |
| LK Burgenlandkreis | [] | [] |
| LK Märkisch-Oderland | [] | [] |
| LK Diepholz | [] | SK Braunschweig, LK Düren |
| LK Forchheim | [] | [] |
| LK Ostholstein | [] | [] |
| LK Osterholz | [] | [] |
| LK Oder-Spree | LK Märkisch-Oderland | [] |
| LK Hameln-Pyrmont | Region Hannover | SK Berlin Friedrichshain-Kreuzberg |
| LK Hochsauerlandkreis | [] | [] |
| LK Ilm-Kreis | [] | [] |
| LK Kitzingen | [] | [] |
| LK Kleve | [] | [] |
| LK Kyffhäuserkreis | [] | [] |
| LK Main-Taunus-Kreis | LK Hochtaunuskreis | [] |
| LK Meißen | [] | [] |
| LK Recklinghausen | [] | [] |

| Target distric state | Detected neighbouring causes | Detected distant causes |
| --- | --- | --- |
| LK Bernkastel-Wittlich | [] | [] |
| LK Neumarkt i.d.OPf. | [] | LK Neustadt a.d.Aisch-Bad Windsheim |
| LK Bad Kreuznach | [] | [] |
| LK Saale-Holzland-Kreis | [] | [] |
| LK Aurich | [] | [] |
| LK Salzlandkreis | [] | [] |
| LK Amberg-Sulzbach | LK Schwandorf | [] |
| LK Saalekreis | [] | [] |
| LK Barnim | [] | [] |
| LK Bayreuth | LK Tirschenreuth | SK Augsburg |
| LK Mansfeld-Südharz | [] | [] |
| LK Lüneburg | [] | LK Konstanz |
| LK Anhalt-Bitterfeld | [] | [] |
| SK Straubing | [] | [] |
| LK Haßberge | [] | [] |
| SK Wuppertal | [] | [] |
| LK Kaiserslautern | [] | [] |
| SK Schwerin | [] | [] |
| LK Holzminden | LK Lippe | [] |
| LK Hof | LK Bayreuth | SK München |
| LK Aschaffenburg | [] | LK Neunkirchen |
| SK Emden | [] | [] |
| LK Mainz-Bingen | [] | [] |
| SK Neustadt a.d.Weinstraße | [] | [] |
| SK Gera | [] | [] |
| SK Oberhausen | [] | [] |
| LK Gifhorn | [] | [] |
| SK Herne | [] | [] |
| SK Salzgitter | [] | [] |
| LK Augsburg | [] | LK Saarlouis |
| SK Kassel | [] | [] |
| SK Kaufbeuren | LK Ostallgäu | [] |
| LK Bad Tölz-Wolfratshausen | LK Weilheim-Schongau | LK Bad Dürkheim, LK Landsberg a.Lech |
| LK Deggendorf | [] | [] |
| SK Ludwigshafen | [] | [] |
| LK Cloppenburg | LK Osnabrück | LK Rhein-Sieg-Kreis |
| LK Börde | [] | [] |
| LK Bamberg | LK Erlangen-Höchstadt, LK Forchheim | SK Berlin Reinickendorf |
| SK Mülheim a.d.Ruhr | [] | [] |
| LK Gütersloh | [] | [] |
| LK Schweinfurt | LK Bad Kissingen, LK Würzburg | SK Ulm |
| SK Cottbus | [] | [] |
| LK Rendsburg-Eckernförde | [] | [] |
| LK Northeim | [] | [] |
| LK Wittmund | [] | [] |
| LK Schleswig-Flensburg | [] | [] |
| LK Uelzen | [] | [] |
| LK Weißenburg-Gunzenhausen | LK Donau-Ries | [] |
| LK Nienburg (Weser) | [] | LK Passau |
| LK Unterallgäu | LK Augsburg, LK Oberallgäu | SK Köln |
| LK Olpe | [] | [] |
| LK Vechta | [] | [] |
| LK Rhein-Lahn-Kreis | [] | [] |
| LK Wetteraukreis | LK Hochtaunuskreis | [] |
| LK Regen | [] | LK Siegen-Wittgenstein |
| LK Cochem-Zell | [] | [] |
| LK Nordsachsen | [] | [] |
| LK Hersfeld-Rotenburg | LK Fulda, LK Vogelsbergkreis | LK Lahn-Dill-Kreis, LK Mettmann |
| LK Berchtesgadener Land | [] | [] |
| LK Potsdam-Mittelmark | [] | [] |
| LK Heidenheim | LK Emmendingen | [] |
| LK Ahrweiler | [] | [] |
| LK Darmstadt-Dieburg | [] | SK Köln |
| SK Landshut | [] | [] |
| LK Südliche Weinstraße | [] | [] |
| LK Nürnberger Land | LK Erlangen-Höchstadt | LK Würzburg, LK Minden-Lübbecke |
| LK Günzburg | LK Alb-Donau-Kreis | SK München |
| LK Göttingen | [] | SK Osnabrück |
| LK Donnersbergkreis | [] | [] |
| SK Hamm | [] | [] |
| LK Freyung-Grafenau | [] | [] |
| LK Dahme-Spreewald | [] | SK Ulm |
| LK Harz | [] | [] |
| LK Schwalm-Eder-Kreis | LK Marburg-Biedenkopf | [] |
| SK Passau | [] | [] |
| LK Schmalkalden-Meiningen | [] | [] |
| LK Altenkirchen | [] | [] |
| SK Bamberg | [] | [] |
| LK Altmarkkreis Salzwedel | [] | [] |
| LK Alzey-Worms | [] | [] |
| LK Miltenberg | LK Aschaffenburg, LK Main-Spessart | LK Bernkastel-Wittlich |
| SK Trier | [] | [] |
| LK Wittenberg | [] | [] |
| LK Eichstätt | LK Donau-Ries | LK Aschaffenburg |
| LK Sankt Wendel | [] | [] |
| LK Schaumburg | [] | [] |
| LK Kusel | [] | [] |
| LK Kulmbach | LK Hof | [] |
| LK Saalfeld-Rudolstadt | [] | [] |
| LK Nordfriesland | [] | [] |

| Target distric state | Detected neighbouring causes | Detected distant causes |
|---|---|---|
| LK Rhön-Grabfeld | LK Schmalkalden-Meiningen | [] |
| LK Rhein-Pfalz-Kreis | [] | [] |
| SK Regensburg | [] | [] |
| LK Zwickau | [] | [] |
| SK Suhl | [] | [] |
| LK Peine | [] | [] |
| SK Memmingen | LK Ravensburg | LK Esslingen, LK Emmendingen |
| LK Eichsfeld | [] | [] |
| LK Steinburg | [] | [] |
| SK Wolfsburg | [] | [] |
| LK Altenburger Land | [] | [] |
| SK Speyer | [] | [] |
| SK Amberg | [] | [] |
| LK Mittelsachsen | [] | [] |
| LK Heidekreis | LK Lüneburg, Region Hannover | SK Nürnberg, LK Heilbronn |
| SK Darmstadt | LK Darmstadt-Dieburg, LK Offenbach | [] |
| LK Erzgebirgskreis | [] | [] |
| LK Helmstedt | SK Braunschweig, SK Wolfsburg | [] |
| LK Nordhausen | [] | [] |
| LK Jerichower Land | [] | [] |
| LK Kronach | LK Kulmbach | LK Berchtesgadener Land |
| LK Lichtenfels | [] | LK Mittelsachsen |
| LK Limburg-Weilburg | [] | [] |
| LK Goslar | [] | [] |
| LK Ludwigslust-Parchim | [] | [] |
| LK Neustadt a.d.Waldnaab | LK Bayreuth, LK Tirschenreuth | [] |
| SK Worms | [] | [] |
| LK Höxter | [] | [] |
| LK Trier-Saarburg | [] | [] |
| LK Neuburg-Schrobenhausen | LK Donau-Ries | SK Frankfurt am Main |
| SK Jena | [] | [] |
| LK Coburg | [] | [] |
| LK Gotha | [] | [] |
| LK Greiz | [] | [] |
| LK Odenwaldkreis | [] | [] |
| LK Wartburgkreis | [] | [] |
| SK Flensburg | [] | [] |
| SK Landau i.d.Pfalz | [] | [] |
| LK Vogtlandkreis | [] | [] |
| LK Ansbach | LK Fürth, LK Roth | LK Dingolfing-Landau, SK Weiden i.d.OPf., LK Erzgebirgskreis |
| SK Ansbach | LK Ansbach | [] |
| SK Brandenburg a.d.Havel | LK Havelland | [] |
| LK Wunsiedel i.Fichtelgebirge | LK Tirschenreuth | SK Halle |
| LK Unstrut-Hainich-Kreis | [] | [] |
| LK Birkenfeld | [] | [] |
| LK Weimarer Land | [] | [] |
| LK Stendal | [] | [] |
| SK Dessau-Roßlau | [] | [] |
| LK Werra-Meißner-Kreis | LK Kassel, LK Schwalm-Eder-Kreis | [] |
| SK Coburg | [] | LK Hochsauerlandkreis |
| LK Nordwestmecklenburg | SK Schwerin | SK Rostock |
| LK Südwestpfalz | [] | [] |
| SK Neumünster | [] | [] |
| SK Potsdam | LK Havelland | SK Erlangen |
| LK Mühldorf a.Inn | LK Landshut, LK Traunstein | [] |
| SK Schweinfurt | [] | [] |
| SK Frankfurt (Oder) | [] | [] |
| LK Prignitz | [] | [] |
| LK Altötting | LK Rottal-Inn, LK Traunstein | SK Heidelberg, LK Hameln-Pyrmont |
| LK Wolfenbüttel | [] | [] |
| LK Uckermark | [] | LK Märkisch-Oderland, LK Rendsburg-Eckernförde |
| SK Bayreuth | [] | LK Neustadt a.d.Waldnaab |
| LK Ostprignitz-Ruppin | LK Oberhavel | [] |
| LK Wesermarsch | [] | [] |
| LK Dillingen a.d.Donau | [] | LK Märkischer Kreis |
| SK Pirmasens | [] | [] |
| LK Sömmerda | [] | [] |
| LK Lüchow-Dannenberg | [] | [] |
| LK Sonneberg | [] | [] |
| LK Hildburghausen | [] | [] |
| SK Zweibrücken | [] | [] |

Here we provide figure 3b enlarged for better visibility.

Figure 8: Detected causal districts for the spread of Covid-19, for each district, using the modified SyPI algorithm. Solid arrows depict causes that are neighbour districts (i.e., sharing a common border). Dashed arrows depict causes that are not. The majority of the detected non-neighbour causes are close to big cities with large airports (MUC, STR, TXL, FDH, FMM, NUE, HAM, FRA, HHN, HAJ, NRN, CGN, DUC, DMT, DRS, BRE, KSF, SCN), and the majority of the detected causes are neighbours to the target. Note that since the dashed arrows are significantly longer than the solid ones, the Figure at first glance seems to show mostly dashed arrows. This is misleading; for a numeric comparison, see Figure 4a. Blue cycles indicate 40km radius around the largest airports. For the district-level analysis, the default thresholds of SyPI were used $(0.01, 0.2)$.

## 9.6 Additional experiments including updated data until 26/09/2020

We update our analysis from the last time's available data, by including four months of daily reported Covid-19 infections per federal state until 26/09/2020.

The updated time series can be seen in Figure 9. We see that until September 26th, there is a second but smaller wave of infections, starting from mid July 2020.

Figure 9: Daily reported Covid-19 infections per federal state until 26/09/2020.

As can be seen in Figure 10 including updated infection time series with four more months of data in our analysis, resulted in detected causes very close to the original detected ones until mid May 2020. Moreover, we observe that the newly detected causes are neighbour states, forming clear clusters of neighbouring causes.

Figure 10: Updating the results including data from four more months since the first analysis, we see that the detected causes are very close to the originally detected ones and more clustered in neighbours.

At this point we attempt to re-evaluate the role of the updated NPIs as well. Since the ending date of each political measure was not always clearly stated for each federal state, we report the following disclaimers. Regarding the banning of visitors on hospitals: from the day that even one visitor was allowed, we considered the measure no longer active on the specific state. Moreover, in federal states where the banning of gatherings of more than 10 people was active, the banning of gathering of more than 2 people was also considered active, even if it was not explicitly stated as a separate measure by the state. Moreover, during the summer holidays that the schools were closed, the measure "closing of schools" was considered active. Since the universities did not open officially for courses, the measure for closing of the universities was considered active. Finally, regarding the banning of gatherings

of more than 1000 people, there seemed to be a difference between public events, private events, events outdoors, events indoors and an exception for the fans for the soccer league. We considered the measure active apart from the states that the attending of soccer in stadiums was allowed.

Taking into account the non pharmaceutical interventions as well, the additional data of four months resulted in the detected causes shown in Figure 11. As we can see, the closing of schools and universities are no longer the most prominent causal policies. Now, the measures most frequently detected as causal are the obligatory quarantine after the visit of risk areas, and the closing of restaurants. In terms of demographics, we observe that less arrows are detected, and that the federal states have causes that include at least one neighbour region. We stress again that cautious should be exercised in the interpretation of these new results, as the termination dates of some of the measures was very vague in most of the states. Therefore, we cannot exclude the possibility that the binary time series that correspond to the NPIs after 15th of May are not correct.

Figure 11: Updated results with infection numbers and NPIs until 26/09/2020, for thresholds (0.05, 0.1).