[Reviews · NeurIPS 2020]

Review 1

Summary and Contributions: The authors present a theorem for causal feature selection for time series data, which they then apply to the spread of COVID-19 in Germany, and the causal impact of restriction measures. They find that their limited dataset contains causal signals that aid analysis of potential interventions. Their main contribution is a theoretical extension to the SyPI algorithm for causal feature selection. SyPI requries that the target be a sink node; the authors relax this assumption, proving that it suffices that none of the descendents of the target belong in the pool of candidate causes (in this application, other regions that have reported COVID-19 cases before the target region).

Strengths: The application highlighted in this paper is very relevant to the COVID-19 pandemic, as the authors aim to determine the impact of restrictions measures to the spread of the disease. Their use of causal analysis is interesting and seems to work quite well in the chosen application. They have presented a novel theoretical extension to the SyPI algorithm.

Weaknesses: The evaluation of the results is mostly qualitative, which is understandable. However, the authors did not mention if they consulted with domain experts (such as epidemiologists) to affirm whether the results are reasonable.

Correctness: The methodology used seems sensible and correct.

Clarity: The paper is well-written and clear.

Relation to Prior Work: The paper does not contain a related works section. They establish the SyPI method and describe their contribution to it, however they do not discuss any related work in using causal analysis in healthcare (or more specifically, epidemiology). It is clear how their work contributes to the field of causal analysis, but it would be meaningful to also include how it contributes to epidemiology - what is the typical analysis used by epidemiologists and policy makers when they determine the potential impact of their policies? Does this method improve upon the practiced methods?

Reproducibility: Yes

Additional Feedback: Update: I would be interested in seeing a discussion section with the authors detailing the feedback they have received from epidemiologists/virologists, as they have noted in the rebuttal. I uphold my original score, but believe that the paper would strongly benefit from such a section.


Review 2

Summary and Contributions: This paper presents a causal analysis of Covid-19 spread in Germany, using SyPI, a causal feature selection method for time series with latent confounders. Given a target region or district in Germany, the analysis aims to find the policies and other regions or districts which might have had causal impact on the spread of Covid-19 of the target region/district. On its theoretical contribution, the paper has relaxed one of the assumptions made by SyPI by allowing the target to be a non-sink node (i.e. to have descendants which are not potential causes of the target).

Strengths: 1. The practical problem addressed and the causal inference methodology involved are relevant to and of great interest to the NeruIPs community. 2. The application of SyPI to causal analysis Covid-19 spread and the direction to relax the assumption of SyPI are both appropriate and promising. 3. The design of the analysis, relative to the limited availability/amount of data, is quite comprehensive, and the findings are meaningful too.

Weaknesses: 1. The main theoretical contribution of the paper is the relaxation of one of the assumptions of SyPI. While the general motivation of removing the sink node assumption (on the target) is obvious, it is not clear, as the paper stands, how this theoretical contribution has helped with the causal analysis of Covid-19 spread presented in the paper. Specifically, in the analysis, the candidate causes input into SyPI were "those other regions that have reported Covid-19 cases BEFORE the target", which sounds that the target was still treated as a sink node without descendants. 2. Although the problem is of great practical significance and the causal approach taken for the analysis can be very useful, the findings from the analysis are more or less not unexpected. The analysis would be more useful if the impacts could be quantified and the data used could be enriched with more factors and more samples included. 3. The theoretical development in the paper is along the promising direction, but overall the theoretical contribution of the paper is quite incremental.

Correctness: The causal approach taken by the paper is correct, and the results, although confined by the small amount of data, have shown the effective of the analysis method to some extent.

Clarity: Overall the paper is clearly written. However, there are some gaps in the presentation, especially regarding the introduction and relation to previous work/theorems/conditions in [6], the original paper of SyPI. For example, readers could refer to [6] for details, but the key parts such as Theorems 1 and 2 (particularly the two conditions) should be presented more formally and explicitly. Another example, how SyPI deals with binary variable, i.e. those policy variables?

Relation to Prior Work: The main related work (theoretically) is [6] which presented the SyPI method. The paper has clearly discussed how their extension is related to SyPI. From application perspective, I understand that there might not be much directly related publications there which had analysed exactly the same problem/data.

Reproducibility: Yes

Additional Feedback: Thanks authors for your response and the clarification on the need of relaxing the sink node assumption. In this case, I think the paper could be strengthened if some experiments (with simulated data for easy validation) are conducted to compare the results with and without the assumption, to demonstrate the benefit of the theoretical extension.


Review 3

Summary and Contributions: The authors extend existing work on necessary and sufficient conditions for causal feature selection weakening the assumption that the target variables cannot have descendants, and instead replacing it with the assumption that none of the target descendants belong to the set of possible causes. This weakening of the assumptions allows the authors to use SyPI to analyze possible causes of COVID-19 infections among German federal states and cities.

Strengths: 1- The authors do an excellent job at summarizing the existing theoretical results, and their extensions on a very intuitive level that allows the reader to follow without getting lost in unnecessary details 2- The authors adequately convey possible limitations of their approach, specifically the possibility of violations of the assumption that all descendants of Y are not contained in X, and possibly non-stationarities of the graph defining the different variables in the COVID data. 3- Both data and code are included with the submission, making it very easy to reproduce their results 4-The authors tackle a timely and important issue of causal analysis of COVID data.

Weaknesses: 1- In several cases there is significant ambiguity (a) Time indices are sometimes dropped without explanations as to which time steps (or maybe these are vectors containing all time steps?). For example line 67, X^i without a _t index, same for line 90. Would the authors be able to clarify? (b) The experiment setup is a bit ambiguous: How do the authors define the set of possible causes and possible targets? are the causes the policies in a given federal state? Or are they the number of COVID cases in a given federal state? What do the authors mean by "we use as candidate causes those other regions that have reported covid-19 cases before the target?" (line 112), again what does it mean for a _region_ to be the cause: is it the number of cases in a given region, or is it the policies enacted in a particular region? Also, what do the authors mean by " reported covid-19 cases before the target?". For example, if state A reported their first case on march 1st, and state B reported their first case on march 2nd, would state A always be considered a cause for state B's infection rates? (c) Finally, it is highly likely that the causes changed over time. For example state A might have caused infections in state B for the first 2 weeks of the pandemic, but then the relationship could have reversed, perhaps because state A enforced strict policies. It is not obvious that the current setting allows for that. Can the authors comment or clarify? 2- The empirical analysis is the main weakness: (a) One set of results that would have really made the authors' point would have been some result on simulated data that shows that whenever none of the target descendants belong to the set of possible causes, recover the true DAG/features. Such a result is relatively common among similar papers (e.g., the Atalanti paper that the authors build upon provide results on simulated data). This kind of result was absent from the paper. Would the authors be able to address why they chose to forgo simulations or if they have simulation results which prove the empirical validity of using SyPI in this setting? (b) It is very hard to read off the maps, especially fig 2b. It is not obvious how to use the results presented on these maps (or elsewhere in the paper) and conclude that SyPI is working as intended here. 3- The authors rightly point out to the fact that some of the assumptions necessary for valid causal reasoning might be violated. In order to gauge the sensitivity of SyPi to different assumptions that are likely going to be broken in a covid analysis, again, results on a simulation seem necessary. Have the authors tested robustness to the "no of the target descendants belong to the set of possible causes" assumption and stationarity assumptions?

Correctness: The theoretical justification seems cogent (I did not check the full proof in the appendix). The empirical results are interesting, but offer limited insights into whether or not the suggested method actually works (see weaknesses)

Clarity: See clarity points outlined in the strengths (clear with respect to the theoretical findings and existing theory), and weakness (ambiguity with respect to the experimental setup and the results)

Relation to Prior Work: Yes. Authors clearly state that the work is building up on recent paper by Atalanti et al.

Reproducibility: Yes

Additional Feedback:


Review 4

Summary and Contributions: The authors propose a new causal inference method, which 1) can address time series data and 2) does not rely on the assumption of causal sufficiency. The new method was applied on Covid-19 case numbers.

Strengths: Casual discovery from time series without requiring causal sufficiency is of great importance in many areas (e.g., healthcare and finance). For this reason, the potential impact of this work is definitely significant. The application (Covid-19) of the proposed method could have profound sociological implications.

Weaknesses: Below are my major concerns. 1. What is the difference between the proposed approach and FCI (and its extensions on time series, tsFCI in reference [5])? 2. Comparing the proposed approach against Lasso-Granger is, in my opinion, not fair, since the latter assumes causal sufficiency. Why not comparing against tsFCI, which allows latent confounders? Please note that the code of tsFCI is publicly available (see the link in the tsFCI paper). 3. I understand the rationale for applying the approach on Covid-19 data. However, since the ground truth is not known, it is very difficult to evaluate the results. Why not also test the method on simulated data where the ground truth is known? This will make the results more convincing.

Correctness: The theorems seem to be correct, although I did not carefully examine them in this paper nor in reference [6], which this paper largely relies on.

Clarity: The paper is very well written.

Relation to Prior Work: Missing the comparison with key prior work (see details in Weaknesses).

Reproducibility: Yes

Additional Feedback: I appreciate the authors' feedback (comparing tsFCI and adding results on simulated data). While these results make the paper stronger, unfortunately I have to see more details to be convinced (I understand that due to space limit, such details cannot be included in the rebuttal). As a result, I would recommend to include these details in future submission (possibly in the supplementary material) so that the reviewers can do a thorough check. For this reason, I will not change my original score.

[Author Response · NeurIPS 2020]

1. Many thanks for the helpful comments, including the "significant impact this method can have on the current Covid-19
2. pandemic", "of great interest to the NeurIPs community" and "The authors do an excellent job at summarizing ... their
3. extensions on a very intuitive level...follow without getting lost". Below we address all reviewers' points and provide
4. **additional simulation** and **comparison experiments**.
5. **R#1:** The method used here provides qualitative instead of quantitative results, in the sense that it can detect the causes
6. of a target even in presence of latent confounders, but not the causal strength. Epidemiological studies in Germany
7. focused on the patient 0 and the first 15 infections in Bavaria, but not on the wider spread. While this leaves us without
8. epidemiological ground truth, we did consult reports from the RKI institute on events that could have contributed to the
9. spread. Since the paper submission, we have presented/discussed the work with virologists/epidemiologists. They were
10. intrigued by the fact that causal inference provides tools that work already on the relatively weak data (case numbers).
11. We will be happy to update the paper to reflect these discussions.
12. **R#2:** 1. Had we not extended the theory to account for non-sink targets, when applying SyPI on them, it would result in
13. less detected causes: Th 1 would still prevent false acceptances, but Th 2 would no longer provide necessary conditions
14. for all unconfounded targets. We relaxed this strict assumption allowing Y to have descendants which do not belong
15. in its candidate causes $\mathbf{X}$. In the Covid-19 dataset, we approach this assumption in practice by assuming that only
16. the infections of the regions that occurred before the target belong to its candidate causes. This way, Y can have
17. descendants in the observed (and unobserved) time series, but not in the subset that contains its candidate causes.
18. 2. We agree that the more factors and samples become available, the more useful the method will be.
19. 3. This is an incremental extension of a theoretical method with a twofold goal: 1. it aims at making the aforementioned
20. method easily applicable to real time series data of the Covid-19 pandemic, so that it can later on be used when more
21. data are available, 2. it provides a causal perspective of the current spread of the pandemic in Germany.
22. Clarifications: In Section 6.2.3 we provide Theorem B, which presents the conditions of Th 1 and 2 combined. The two
23. theorems of SyPI, and the proposed extension theorem do not depend on the type of the variable (e.g., binary or not).
24. **R#3:** 1-a: In l 67 and 90 we deliberately omit the "$_t$" index to denote the whole time series, which is in line with our
25. notation in l 59-60. 1-b: As we explain in Section 2.4 we ran two different experiments: first at the federal state level,
26. and then at the district level. In both cases the time series are the daily reported Covid-19 infections in a (federal
27. state or district) region. We assign every time one regional infection series to be the target Y, and all the remaining
28. regional time-series that have reported infections before the target to be the pool of its candidate causes, from which
29. SyPI will then identify the true causes. As we explain, we do this to comply with our proposed modified assumption
30. $\mathbf{DE}_{\mathbf{Y}}^{\mathcal{G}} \notin \mathbf{X}$. For the federal state-level analysis, in addition to the regional infection series, we use as candidate causes
31. the binary time series of the policies that were applied in the target state. The fact that a time series is assigned in the
32. pool of candidate causes does not mean that it will indeed be a true cause. This is what the proposed method identifies.
33. Therefore, the correct phrasing would be "if state A reported on 1/3 and state B on 2/3, then the daily infections of A
34. will be used as a candidate cause of target B, and then SyPI will identify if indeed A causes B". 1-c: We discuss this
35. case in the last par. of Section 2.2. As we mention, SyPI relies on the stationarity of the causal relationships in the
36. graph. If this is violated (i.e. it could be that the policies not only cause the reported infection time series but also be
37. caused by it in different time windows), then the method will no longer correctly detect the causes.
38. 2-a: Please see added simulations in point 3 below. 2-b: As there is no ground truth, we can only provide evidence that
39. our results seem meaningful. This is why in Section 4.1 we provide (admittedly limited) information about the location
40. of the detected causes with respect to airports and major events that took place, as well as comparison of our findings
41. about the role of the political interventions with other methods which used similar dataset [9].
42. 3: For the potential violation of the proposed assumption please see answer to point 1. of R#2.
43. **R#4:** 1. The main difference between the proposed approach and tsFCI is that SyPI pre-calculates a very concise
44. conditioning set for each target and only requires two conditional independence (CI) tests per candidate cause, to decide
45. if it is a true cause of the target. In contrast, tsFCI performs exhaustively CI tests for all possible combinations of
46. conditioning sets and lags, which results in very ambiguous statistical results and very large computational times in
47. large graphs. Of course, tsFCI aims at the full graph discovery and not only at causal feature selection (SyPI). This also
48. justifies tsFCI's more computationally intensive conditions.
49. 2. As was requested, we performed **additional comparisons with tsFCI** for the infections in the federal states.
50. For fair comparison we used the same threshold for all the statistical tests of both methods (0.05). Due to lack of
51. space here we describe the results and we will add the figure in the manuscript. tsFCI detected 8, while SyPI 44
52. directed edges (causes). 4 of the tsFCI were a subset of the ones detected by SyPI. For the majority of the remaining
53. states tsFCI yielded '↔'. SyPI needed only 19 seconds to run, while tsFCI needed 15 minutes for the same dataset.
54.

3. As requested, here we provide 33 **experiments on simulated graphs** (100 graphs/experiment) with 1000 samples, varying noise and number of time-series, with two hidden, allowing the target to have descendants that do not belong to its candidate causes. The FNR for direct causes (dashed) remains below $40\%$ as in [6], and the FPR (continuous) is close to 0. As expected from the proof, SyPI's performance was not affected.

[Meta-Review · NeurIPS 2020]

As noted by the reviewers, there are promising developments in this paper, and timely and interesting applied results. The technical quality can be further improved, however, given the extraordinary scientific challenges posed by the pandemic, I would argue that the paper would nevertheless be quite welcome in a dedicated COVID-19 track. Some suggestions for the authors, hopefully useful as they revise it: 1) while Germany had better case reporting then other countries, nevertheless, especially in the early epidemic, cases were severely underreported as compared to the true number of infections. Thus, the focus on case data, and on temporal ordering based on which state/district reported cases first, is a potential shortcoming. This could be addressed by focusing on deaths data, at least as a sensitivity analysis. 2) data on mobility is available from a number of different sources; this would provide a direct way to test the conclusions made by the SyPI about movement from one region to another. 3) the time period covered was until mid-May; updating the paper to include more recent data, and thus the impact of the relaxation of NPIs, would be very interesting. Also this would enable you to evaluate the predictions you made, by seeing whether the relaxation of any of the important NPIs you highlighted led to a resurgence in cases.